# Highly Selective Gas Sensor Based on Litchi-like g-C_3_N_4_/In_2_O_3_ for Rapid Detection of H_2_

**DOI:** 10.3390/s23010148

**Published:** 2022-12-23

**Authors:** Ji Zhang, Xu Li, Qinhe Pan, Tong Liu, Qingji Wang

**Affiliations:** 1State Key Laboratory of Marine Resource Utilization in South China Sea, College of Information and Communication Engineering, Hainan University, Haikou 570228, China; 2School of Chemical Engineering & Light Industry, Guangdong University of Technology, Guangzhou 510006, China; 3School of Chemical Engineering and Technology, Hainan University, Haikou 570228, China; 4School of Electronic and Information Engineering, Qingdao University, 308 Ningxia Street, Qingdao 266071, China

**Keywords:** g-C_3_N_4_/In_2_O_3_, litchi-like, H_2_ sensor, fast response, high selectivity

## Abstract

Hydrogen (H_2_) has gradually become a substitute for traditional energy, but its potential danger cannot be ignored. In this study, litchi-like g-C_3_N_4_/In_2_O_3_ composites were synthesized by a hydrothermal method and used to develop H_2_ sensors. The morphology characteristics and chemical composition of the samples were characterized to analyze the gas-sensing properties. Meanwhile, a series of sensors were tested to evaluate the gas-sensing performance. Among these sensors, the sensor based on the 3 wt% g-C_3_N_4_/In_2_O_3_ (the mass ratio of g-C_3_N_4_ to In_2_O_3_ is 3:100) showeds good response properties to H_2_, exhibiting fast response/recovery time and excellent selectivity to H_2_. The improvement in the gas-sensing performance may be related to the special morphology, the oxygen state and the g-C_3_N_4_/In_2_O_3_ heterojunction. To sum up, a sensor based on 3 wt% g-C_3_N_4_/In_2_O_3_ exhibits preeminent performance for H_2_ with high sensitivity, fast response, and excellent selectivity.

## 1. Introduction

Hydrogen (H_2_) is a potential clean energy that has rich application prospects in automobile, aerospace, and other fields [1]. However, due to its flammable and explosive characteristics, H_2_ can very easily cause disaster during utilization and storage [2]. Therefore, an effective survey of H_2_ has become an important problem to be solved. At present, detection methods of H_2_ include gas chromatography, mass spectrometry, gas sensors, and so on [3]. In the above methods, gas sensors have become the best choice, with the advantages of low cost, simple preparation, and small size [4]. Furthermore, the type of gas sensor include semiconductor oxide [5], catalytic combustion [6], solid electrolyte [7,8], etc. Among them, a semiconductor oxide sensor is widely used in people’s lives because of the outstanding performance [9]. So far, many semiconductor oxides have been researched as sensing-materials in H_2_ sensors, such as TiO_2_, ZnO, SnO_2_, NiO, and In_2_O_3_. Some reported H_2_ sensors based on different semiconductor oxides are summarized in Table 1.

In_2_O_3_ is a common semiconductor with a band gap of 3.6 eV, which is broadly used in gas sensors [10]. Up to now, preparing In_2_O_3_ with distinct morphologies has proved to be an available measure to heighten the performance of gas sensors. However, a single In_2_O_3_-based sensor has the defects of low response and poor selectivity in practical applications [11]. Many researchers have made great efforts to enhance the gas-sensing performance, such as doping noble metals [12], fabricating heterojunctions [13], and constructing 2D materials or polymer composites [14,15]. Graphitic carbon nitride (g-C_3_N_4_) is a typical polymer semiconductor with a band gap of 2.76 eV, which has been used in gas sensors owing to high chemical stability, large surface area, and good catalytic function [16,17]. It is reported that In_2_O_3_-based sensors can elevate gas-sensing performance through the g-C_3_N_4_ composite. Liu et al. prepared In_2_O_3_, and g-C_3_N_4_ was compounded by way of MOF, and the response value of the sensor achieved 294 to 100 ppm NO_x_ at RT [18]. Sun et al. synthesized g-C_3_N_4_/In_2_O_3_ through a calcination annealing process, and the response value of the sensor reached 1405 at 119 °C to 100 ppm formaldehyde [19]. The enhanced performance of the In_2_O_3_ sensor can be attributed to the formation of g-C_3_N_4_/In_2_O_3_ heterojunction different oxygen species content [20].
sensors-23-00148-t001_Table 1Table 1Gas-sensing properties of H_2_ sensor made by some semiconductor oxides.Sensing MaterialsConc. (ppm)Res.τ_res._ (s)τ_rec._ (s)Ref.WO_3_-TiO_2_100005.62 ^c^485[21]Ag/ZnO300479% ^b^175655[22]Pd/SnO_2_10001.2 ^a^21451.5[23]Pt@NiO50004.25 ^c^918[24]Pd-doped In_2_O_3_1003.6 ^a^47[25]**3 wt% g-C_3_N_4_/In_2_O_3_****100****180% ^b^****2****2.4****This work**Conc.: Concentration; Res.: Response; τ_res._: Response time; τ_rec._: Recovery time; Ref.: reference; ^a^ R = R_a_/R_g_; ^b^ R = (R_a_ − R_g_)/R_g_ × 100%, ^c^ R = R_g_/R_a_.

In this paper, a litchi-like g-C_3_N_4_/In_2_O_3_ composite was successfully prepared by the hydrothermal method. Additionally, the morphology and composition of C_3_N_4_/In_2_O_3_ were characterized by XRD, SEM, TEM, and XPS. Sensors based on different amount of g-C_3_N_4_ composite were fabricated to investigate their gas-sensing specifics. Among them, the performance of the sensor based on 3 wt% g-C_3_N_4_/In_2_O_3_ was significantly improved for H_2_, giving it the merits of a fast response and excellent selectivity.

## 2. Experiment

### 2.1. Synthesis of Peachcore-like Pure In_2_O_3_

All chemicals are purchased from Aladdin Reagent and are analytical grade without being further purified for use. In a typical process, 147.5 mg of InCl_3_·4H_2_O was dissolved in 15 mL of deionized water, then 15 mL of glycerol was added and stirred until it was homogeneous. Afterwards, 520 mg of Na_3_C_6_H_5_O_7_·2H_2_O was added into the above solution and stirred vigorously for 20 min. Finally, 250 μL of NaOH (0.1 M) aqueous solution was added slowly to form a uniform solution. The obtained solution was transferred to an autoclave lined with 50 mL PTFE for a hydrothermal reaction at 190 °C for 16 h. After being cooled to room temperature, the precipitate was centrifuged with deionized water and absolute ethanol for several times. The powder was collected and dried at 80 °C overnight. The dried sample was placed in an Al_2_O_3_ boat and calcined in a muffle furnace at 400 °C for 2 h (2 °C/min) to obtain peachcore-like pure In_2_O_3_.

### 2.2. Synthesis of Litchi-like g-C_3_N_4_/In_2_O_3_

Firstly, 20 g of urea was added into a lidded crucible and heated in a muffle furnace at 550 °C for 2 h (2 °C/min) to prepare g-C_3_N_4_. Then, the prepared g-C_3_N_4_ was dispersed in 15 mL deionized water and sonicated for 2 h to ensure dispersion. Subsequently, 147.5 mg of InCl_3_·4H_2_O and 15 mL of glycerol was added the above solution in turn and stirred until homogeneous. Thereafter, 520 mg of Na_3_C_6_H_5_O_7_·2H_2_O was added and stirred vigorously for 20 min. Finally, 250 μL of NaOH (0.1 M) aqueous solution was added to form a uniform solution. The subsequent process was the same as the preparation of pure In_2_O_3_. According to the different contents of g-C_3_N_4_, 1 wt% g-C_3_N_4_/In_2_O_3_, 3 wt% g-C_3_N_4_/In_2_O_3_, and 5 wt% g-C_3_N_4_/In_2_O_3_ composites were prepared, respectively.

### 2.3. Material Characterization

The crystal structures of as-samples were determined by an X-ray diffractometer (XRD, Rigaku Miniflex 600 X, Cu Kα1 radiation, λ = 1.5406 Å) operated at 40 kV and 15 mA. The morphology characteristics of samples were characterized by a scanning electron microscope (SEM, PHENOM SCIENTIFIC ProX G5, The Netherlands) and a transmission electron microscope (TEM, FEI Tecnai G2 F30, USA). X-ray photoelectron spectroscopy (XPS, Thermo escalab 250Xi, USA) was used for the chemical composition analysis of samples.

### 2.4. Fabrication and Measurement of Gas Sensors

The device structure of the gas sensor is shown in Figure 1. As-prepared samples were mixed with deionized water to form a uniform slurry and coated on an alumina ceramic tube as the sensing layer. The coated sensing layer was baked under an infrared lamp for 15 min. Then, the ceramic tube was calcined at 300 °C for 1 h (2 °C/min). Finally, the heating wire was passed through the ceramic tube and welded to a six-legged base to make a gas sensor. In addition, the components of the test gas are the target gas and the component gas, wherein the standard value of the target gas is 1% mol/mol and that of the component gas is nitrogen.

The evaluation of gas-sensing performances was carried out in a static test system (50% RH, 25 °C), as shown in Figure 2. The heating current of the sensor was provided by the DC-regulated power supply, and the resistance value was recorded by the multimeter. The response value is defined as R = (R_a_ − R_g_)/R_g_ × 100%, wherein R_a_ and R_g_ are the resistance value of the sensor respectively exposed to clean air and target gas. Additionally, the response/recovery time is defined as the time for 90% of the resistance change. 

## 3. Results and Discussion

### 3.1. Characterization of Material Structure

The crystal phase of the as-prepared samples was obtained by XRD, as shown in Figure 3. The g-C_3_N_4_ has two peaks at 2θ angles of 13.36° and 27.38°, which are related to the tris-triazine units and aromatic systems, respectively [26]. The diffraction peaks of pure In_2_O_3_ at 2θ angles of 22.37°, 30.99°, 32.61°, 45.61°, 50.25°, 57.20° and 58.19° are index to the crystal planes (012), (104), (110), (024), (116), (214) and (300) of In_2_O_3_ (JCPDS 22-0366). However, the diffraction peaks of g-C_3_N_4_ are not obviously observed in the XRD pictures of g-C_3_N_4_/In_2_O_3_ composites, probably due to the low content of g-C_3_N_4_ [27]. 

The SEM images of pure In_2_O_3_ and 3 wt% g-C_3_N_4_/In_2_O_3_ are shown in Figure 4. Pure In_2_O_3_ exhibits a peachcore-like structure with a diameter of about 250 nm in Figure 4a,b. The morphology of 3 wt% g-C_3_N_4_/In_2_O_3_ possesses a distinctive litchi-like structure, and the diameter of 3 wt% g-C_3_N_4_/In_2_O_3_ is only 130 nm, as shown in Figure 4c,d. Apparently, the morphology of In_2_O_3_ was changed from peachcore-like to litchi-like, and the diameter of 3 wt% g-C_3_N_4_/In_2_O_3_ is smaller about 100 nm than that of pure In_2_O_3_. Moreover, the size of 3 wt% g-C_3_N_4_/In_2_O_3_ is more uniform than that of pure In_2_O_3_. The crystallite sizes of as-prepared samples were calculated by the Scherrer formula:(1)$$D=\frac{K\lambda }{\beta cos\theta }$$
where *K* is Scherrer constant of 0.9, λ is the X-ray wavelength of 0.15406 nm, β is the half-width of the diffraction peak, and θ is the Bragg diffraction angle. The average size of pure In_2_O_3_, 1 wt% g-C_3_N_4_/In_2_O_3_, 3 wt% g-C_3_N_4_/In_2_O_3_, and 3 wt% g-C_3_N_4_/In_2_O_3_ is about 15 nm, 14.4 nm, 13.9 nm and 12.6 nm, respectively. Based on the above results, it is inferred that the g-C_3_N_4_ may inhibit the growth of In_2_O_3_ crystal, thus leading to morphological changes [28].

The microstructure of 3 wt% g-C_3_N_4_/In_2_O_3_ was characterized by TEM and HRTEM, as shown in Figure 5. Through TEM in Figure 5a, we can clearly see that the 3 wt% g-C_3_N_4_/In_2_O_3_ presents a litchi-like structure with uniform size. Furthermore, HRTEM in Figure 5b is manifested by the different lattice spacings of In_2_O_3_, which plainly indicates In_2_O_3_ with a highly crystalline form. The three lattice spacings are 0.12 nm, 0.18 nm and 0.17 nm, which are assigned to the (012), (110) and (104) planes of In_2_O_3_, respectively. In addition, the element mapping is shown in Figure 5c–h. From element mapping, it can be explicitly noticed that the In, O, C, and N elements are evenly distributed; this can be evidence that the 3 wt% g-C_3_N_4_/In_2_O_3_ composite was successfully prepared.

To analyze the chemical composition of pure In_2_O_3_ and 3 wt% g-C_3_N_4_/In_2_O_3_, the XPS is shown in Figure 6. As shown in Figure 6a, the In 3d of pure In_2_O_3_ and 3 wt% g-C_3_N_4_/In_2_O_3_ have two strong peaks at 451.6 eV, 444.1 eV and 451.7 eV, 444.2 eV, which correspond to In 3d_5/2_ and In 3d_3/2_, respectively [29]. The O 1s peak spectrums in Figure 6b are decomposed into three fitting peaks around 532.1 eV, 531.2 eV, 529.5 eV and 532.2 eV, 531.2 eV, 529.6 eV, wherein the fitting peaks are assigned to the hydroxyl (OH) or chemisorbed oxygen (O_C_), oxygen vacancy (O_V_), and lattice oxygen (O_L_) [30,31]. Furthermore, the O_C_, O_V_, and O_L_ content of pure In_2_O_3_ and 3 wt% g-C_3_N_4_/In_2_O_3_ are about 8.56%, 23.59%, 67.85% and 9.14%, 33.42%, 57.44%, respectively. It is worth noting that the content of O_C_ and O_V_ increases with the introduction of g-C_3_N_4_. It may be one of the reasons why the gas-sensing performance of 3wt% g-C_3_N_4_/In_2_O_3_-based sensors has improved [32,33]. The C 1s spectrum displays three peaks at 288.3 eV, 286.3 eV, and 284.8 eV, as shown in Figure 6c. These three peaks of C 1s respectively belong to the sp^2-^ bonded carbon (N–C=N) and the sp^3-^ fitted carbon bond from surface defects of g-C_3_N_4_ and carbon atoms (C-C) [34]. However, the N 1s peak appears to be a weak peak at 398.8eV, as shown in Figure 6d, which may be caused by the low content of g-C_3_N_4_ [35].

### 3.2. Gas-Sensing Properties

To determine the optimal operating temperature of the sensors, the different sensors based on g-C_3_N_4_/In_2_O_3_ composites with different proportions were evaluated from 225 °C to 300 °C, as shown in Figure 7. When the operating temperature reaches 275 °C, the response value is 180% to 100 ppm H_2_ of 3 wt% g-C_3_N_4_/In_2_O_3_ sensor in Figure 7a, which is 3.5 times that of pure In_2_O_3_ sensor. The operating temperature can be explained according to the desorption equilibrium of the gas molecules and the chemical reaction kinetics. When the working temperature is too low, the gas molecules do not have enough heat energy and kinetic energy to react on the In_2_O_3_ surface, so the adsorption capacity of the gas is reduced. However, when the operating temperature is too high, the gas molecules adsorbed on the In_2_O_3_ surface will have a high activity and escape before the electron carrier transfer, resulting in a reduced response [3,36]. In addition, the response values of different sensors were tested with different concentrations of H_2_ (10~1000 ppm) at 275 °C. As exhibited in Figure 7b, the response value of the 3 wt% g-C_3_N_4_/In_2_O_3_ sensor is greatly improved compared with other sensors within the whole range of H_2_ concentrations. With the increase in the H_2_ concentration, the rising trend of the response value curve is gradually stable, which may indicate that the sensor is gradually saturated.

The response/recovery time of the different sensors at 275 °C is shown in Figure 8a–d. It is not hard to notice that the increase of g-C_3_N_4_ content, which may have a positive action in the sensor. Although g-C_3_N_4_ is beneficial to the response and recovery characteristics of the sensor, excessive g-C_3_N_4_ may play a reverse role. Furthermore, the 3 wt% g-C_3_N_4_/In_2_O_3_ sensor exhibits fast response/recovery time (2 s/2.4 s) to 100 ppm H_2_. From Figure 8e, even if it is in a high concentration H_2_ atmosphere, the 3 wt% g-C_3_N_4_/In_2_O_3_ sensor also demonstrated extremely fast response/recovery time. The above results reveal that the 3 wt% g-C_3_N_4_/In_2_O_3_ sensor has great potential in practical application. The R_a_ of different sensors at different temperatures are shown in Figure 8f; the R_a_ decrease with the increase in temperature. In addition, the R_a_ of sensors based on g-C_3_N_4_/In_2_O_3_ is lower than that of the pure In_2_O_3_ and decreased with the increasing amount of g-C_3_N_4_ in the composites. The reason for the increase in conductivity is firstly due to the increase in oxygen vacancy in g-C_3_N_4_/In_2_O_3_. The second reason is attributed to the C–N bond breaking, which releases a lot of electrons [37]. Based on the above reasons, it can be proven that the sensor based on g-C_3_N_4_/In_2_O_3_ has high conductivity.

Stability and reproducibility are important indicators of a sensor’s performance, and the evaluation result of the 3 wt% g-C_3_N_4_/In_2_O_3_ sensor at 275 °C is shown in Figure 9. As shown in Figure 9a, the dynamic curves of the 3 wt% g-C_3_N_4_/In_2_O_3_ sensor is exhibited. Within the variation range of H_2_ concentrations, the response and recovery characteristics of the 3 wt% g-C_3_N_4_/In_2_O_3_ sensor are stable. Meanwhile, the detection limit was tested to 1 ppm H_2_, and the response value is 10%. Reproducibility was evaluated by continuous exposure to 50 ppm H_2_ for five cycles, as shown in Figure 9b. In the five-cycle experiment, the response/recovery time and response value were basically stable without significant change. By assessing the gas-sensing properties of the 3 wt% g-C_3_N_4_/In_2_O_3_ sensor, it was sufficienly proven that the sensor has fine stability and reproducibility.

Selectivity is a momentous parameter for the practical application of the sensor. Various gases were tested at 275 °C for the appraisement of different sensors, as presented in Figure 10. Among them, 1 wt% g-C_3_N_4_/In_2_O_3_ and 3 wt% g-C_3_N_4_/In_2_O_3_ sensors exhibit similar selectivity characteristics, which possess a higher response value to 50 ppm H_2_ than other gases (50 ppm SO_2_, 50 ppm NH_3_, 50 ppm CO_2_, 50 ppm CO, 50 ppm CH_4_, 100 ppm NO_2_). Besides, the response value of the 3 wt% g-C_3_N_4_/In_2_O_3_ sensor is three times that of the 1 wt% g-C_3_N_4_/In_2_O_3_ sensor to 50 ppm H_2_. Meanwhile, the response value of the 3 wt% g-C_3_N_4_/In_2_O_3_ sensor to 50 ppm H_2_ is 7.5 times that of 50 ppm SO_2_ (the secondary response gas). As we all know, CO, CH_4_, and H_2_ are common fuel gases, and the response values of the 3 wt% g-C_3_N_4_/In_2_O_3_ sensor to H_2_ are 20 times and 40 times that of CO and CH_4_ of 50 ppm, respectively. The improvement of H_2_ selectivity is due to the better dispersion of litchi-like 3wt% g-C_3_N_4_/In_2_O_3_ compared with pure In_2_O_3_, which makes H_2_ with the smallest molecular size easy to diffuse [38]. In addition, at 275 °C for the sensor based on the 3 wt% g-C_3_N_4_/In_2_O_3_ surface, the adsorption energy of H_2_ gas is much higher than that of other gases [22]. From the above results, the 3 wt% g-C_3_N_4_/In_2_O_3_ sensor exhibits excellent selectivity in H_2_ detection.

The long-term stability is shown in Figure 11, which was evaluated by continuous exposure to 50 ppm H_2_ at 275 °C within 2 weeks. It is distinctly perceived from Figure 11a that the response value to 50 ppm H_2_ is not significantly changed. Moreover, the transient curves for different days are shown in Figure 11b–d, and it is noteworthy that the air resistance (R_a_) and response features of the sensor are basically stable. In view of the above assessment, it can be assured that the 3 wt% g-C_3_N_4_/In_2_O_3_ sensor has good long-term stability for H_2_ detection.

### 3.3. Gas Sensing Mechanism

In order to understand the enhanced properties of the H_2_ sensor based on g-C_3_N_4_/In_2_O_3_, the gas-sensing mechanism of the In_2_O_3_ sensor was analyzed. In air at different temperatures, the oxygen molecules on the surface of In_2_O_3_ materials will become O_2_^-^ (<147 °C), O^−^ (147 °C–397 °C), O^2-^ (>397 °C), as follows in Equations (2)–(5) [29].
O_2_ → O_2(ads)_(2)
O_2(ads)_ + e^−^ → O_2_^−^_(ads)_(3)
O_2_^−^_(ads)_ + e^−^ → 2O^−^_(ads)_(4)
O^−^_(ads)_ + e^−^ → O^2−^_(ads)_(5)
2 H + O^−^_(ads)_ → H_2_O + e^−^(6)

In this paper, the optimum operating temperature was determined to be 275°C, so most of the oxygen molecules are converted into O^−^. When the In_2_O_3_ sensor exposure to H_2_, the H molecules are oxidized by O^−^ to form H_2_O and free electrons as in Equation (6) [22]. The reaction process is shown in Figure 12a. The elevated gas-sensing performance of sensor based on 3 wt% g-C_3_N_4_/In_2_O_3_ may be attributed to the morphology characteristic, the oxygen state and the g-C_3_N_4_/In_2_O_3_ heterojunctions. The first reason is the distinctive litchi-like morphology of 3 wt% g-C_3_N_4_/In_2_O_3_. Through observing SEM and TEM photos, the 3 wt% g-C_3_N_4_/In_2_O_3_ has smaller size and better dispersion than pure In_2_O_3_. The performance of the sensor based on 3 wt% g-C_3_N_4_/In_2_O_3_ may be improved due to the unique morphology configuration.

The second reason can be attributed to the adjustment of the oxygen state. The XPS results show that the O_V_ and O_C_ contents in 3 wt% g-C_3_N_4_/In_2_O_3_ are higher than pure in In_2_O_3_. For the sensor based on 3 wt% g-C_3_N_4_/In_2_O_3_, a large amount of O_V_ provides more active sites for the adsorption of active oxygen, and the increase in O_C_ content indicates that more chemically adsorbed oxygen participates in the redox reaction [19]. Therefore the performance of the 3 wt% g-C_3_N_4_/In_2_O_3_ sensor may be improved. The last reason is the formation of the g-C_3_N_4_/In_2_O_3_ heterojunction [19]. The work functions and band gaps of g-C_3_N_4_ and In_2_O_3_ are W = 4.3 [39], E_g_ = 2.76 eV [16] and W = 5.0 [13], E_g_ = 3.6 eV [10], respectively, as exhibited in Figure 12b. The electrons flow from g-C_3_N_4_ to In_2_O_3_ to the new equilibrium of the Fermi level. Therefore, the sensor based on 3 wt% g-C_3_N_4_/In_2_O_3_ shows excellent response characteristics [40].

## 4. Conclusions

In this work, the g-C_3_N_4_/In_2_O_3_ composite was prepared by a hydrothermal method. The morphology features and chemical compositions of g-C_3_N_4_/In_2_O_3_ were characterized by XRD, SEM, TEM and XPS. The sensor based on 3 wt% g-C_3_N_4_/In_2_O_3_ perform excellent gas-sensing behavior. The response value of 3 wt% g-C_3_N_4_/In_2_O_3_ was 180% to 100 ppm H_2_ at 275 °C, which is 3.5 times higher than that of the pure In_2_O_3_ sensor. Furthermore, the 3 wt% g-C_3_N_4_/In_2_O_3_ sensor exhibits fast response/recovery time (2 s/2.4 s) to 100 ppm H_2_ and excellent selectivity (R_50 ppm H2_/R_50 ppm CH4_ = 40, R_50 ppm H2_/R_50 ppm CO_ = 20). The improvement of sensor performance based on 3 wt% g-C_3_N_4_/In_2_O_3_ can be attributed to the special morphology characteristic, the state of oxygen, and the g-C_3_N_4_/In_2_O_3_ heterojunctions. In conclusion, this work provides a useful composite for preparing highly efficient H_2_ sensors and proves that this composite has certain application value.

## Figures and Tables

**Figure 1 sensors-23-00148-f001:**
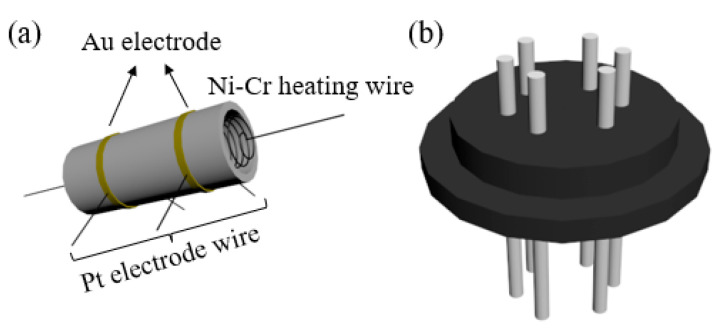
Sensor device structure (**a**) ceramic tube; (**b**) tube base.

**Figure 2 sensors-23-00148-f002:**
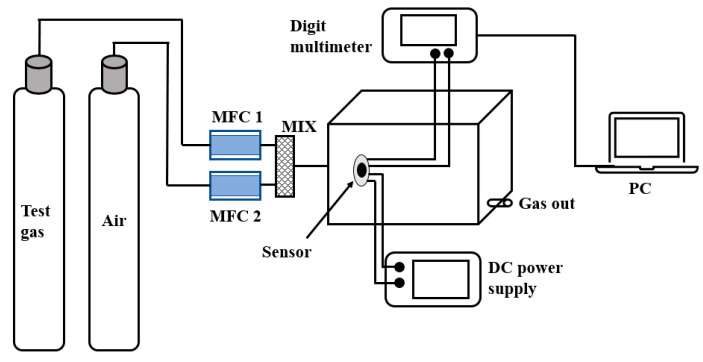
The gas-sensing test system.

**Figure 3 sensors-23-00148-f003:**
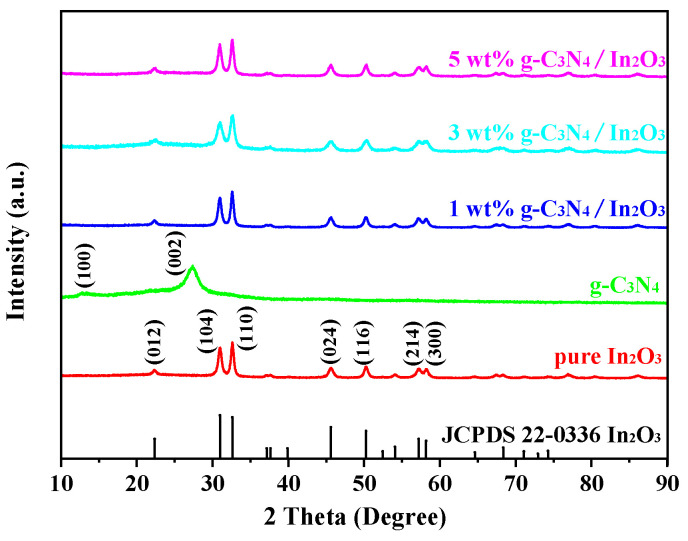
XRD patterns of as-prepared samples.

**Figure 4 sensors-23-00148-f004:**
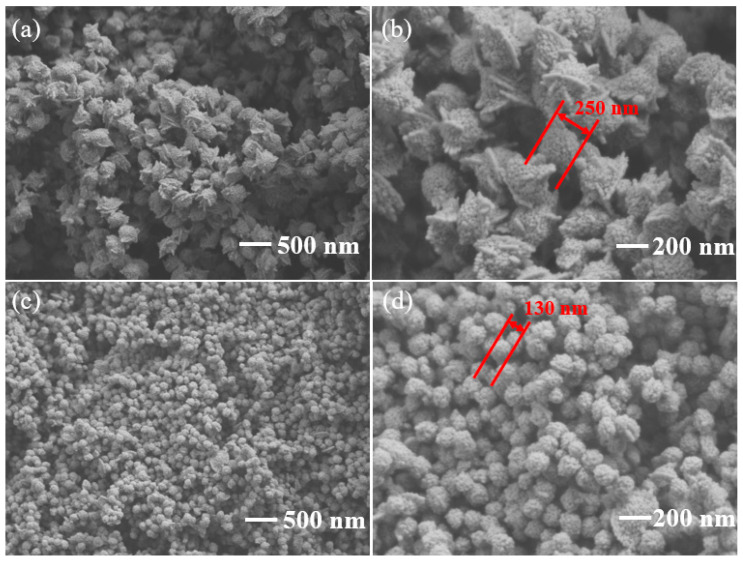
SEM images of (**a**,**b**) pure In_2_O_3_; (**c**,**d**) 3 wt% g-C_3_N_4_/In_2_O_3_.

**Figure 5 sensors-23-00148-f005:**
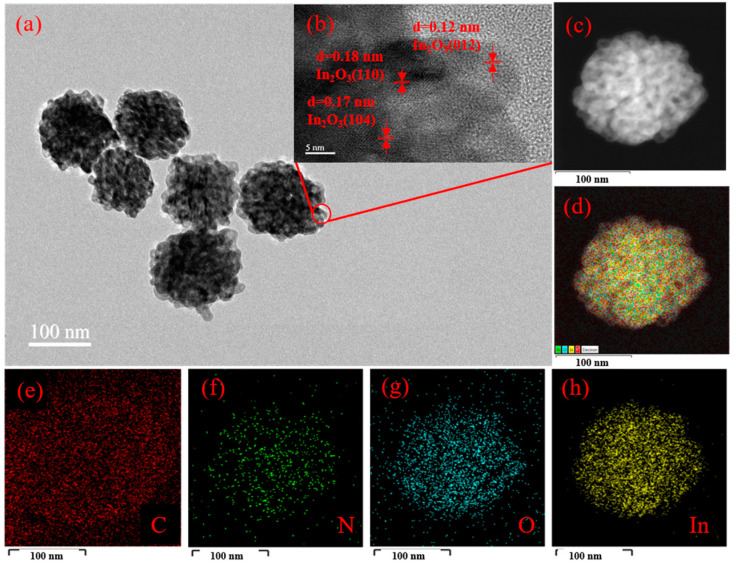
3 wt% g-C_3_N_4_/In_2_O_3_ (**a**) TEM image; (**b**) HRTEM; (**c**–**h**) elemental mapping.

**Figure 6 sensors-23-00148-f006:**
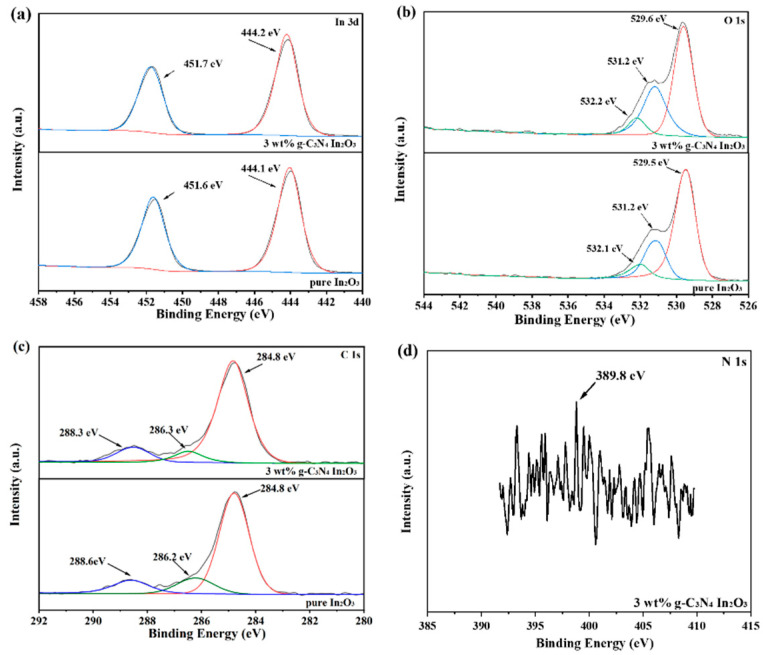
XPS spectra of as-prepared samples (**a**) In 3d; (**b**) O 1s; (**c**) C 1s; (**d**) N 1s.

**Figure 7 sensors-23-00148-f007:**
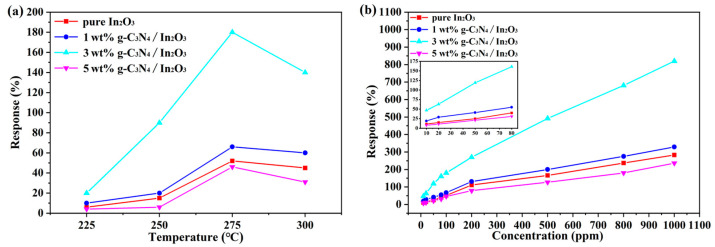
(**a**) Response values of sensors vs. operating temperatures to 100 ppm H_2_; (**b**) response values of sensors vs. different concentrations H_2_ at 275 °C.

**Figure 8 sensors-23-00148-f008:**
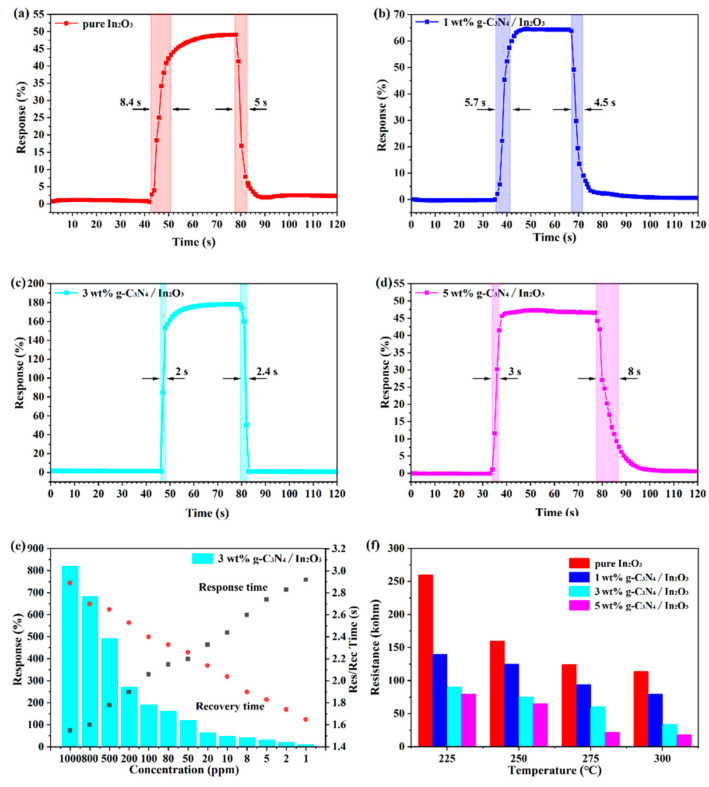
(**a**–**d**) Response/recovery time of different sensors; (**e**) response/recovery time of 3 wt% g-C_3_N_4_/In_2_O_3_ to different concentrations of H_2_ at 275 °C; (**f**) R_a_ at different temperatures of different sensors.

**Figure 9 sensors-23-00148-f009:**
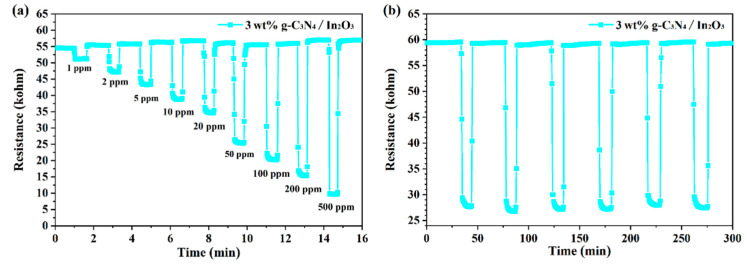
(**a**) Dynamic response curve of sensor to different concentrations of H_2_ at 275 °C; (**b**) the five-cycle response/recovery curve of the sensor to 50 ppm H_2_ at 275 °C.

**Figure 10 sensors-23-00148-f010:**
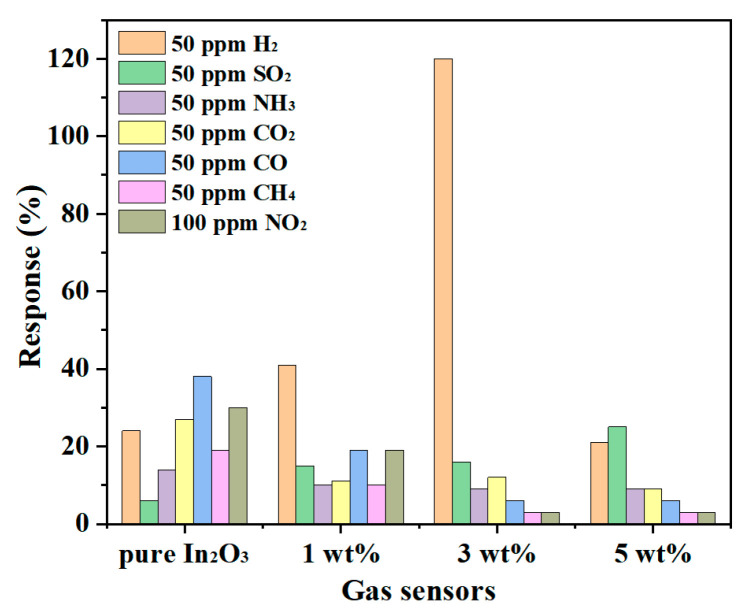
Gas selectivity of different sensors at 275 °C.

**Figure 11 sensors-23-00148-f011:**
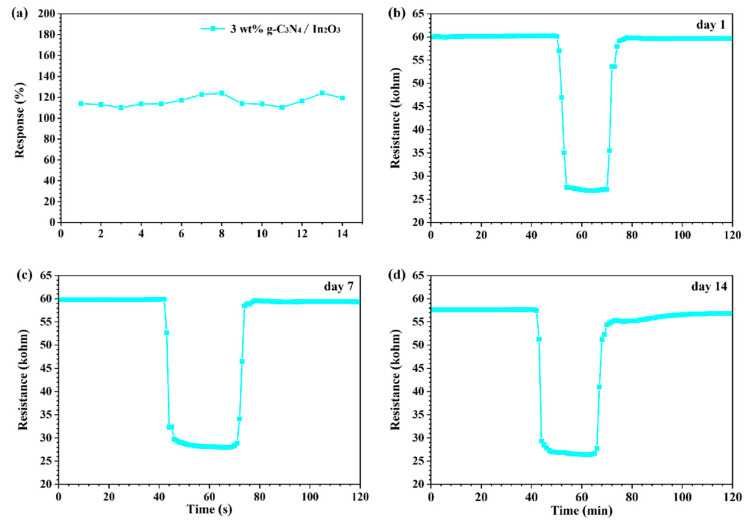
(**a**) Long-term stability over 2 weeks for the sensor to 50 ppm H_2_ at 275 °C; (**b**–**d**) transient curve of different days.

**Figure 12 sensors-23-00148-f012:**
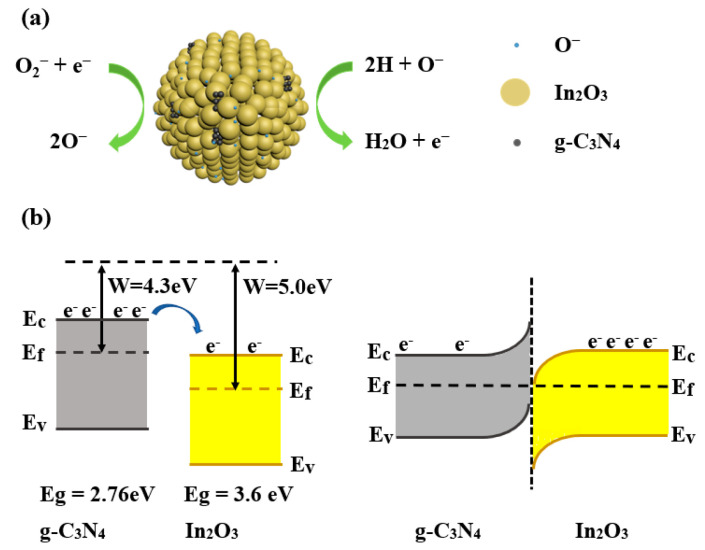
(**a**) Sketch map of gas-sensing mechanism; (**b**) energy band diagram of In_2_O_3_ and g-C_3_N_4_.

## Data Availability

The data of this study can be obtained from the corresponding authors.

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
