# Peer review of "Highly Selective Gas Sensor Based on Litchi-like g-C3N4/In2O3 for Rapid Detection of H2"

_sensors, 2022, doi:10.3390/s23010148_

Round 1

Reviewer 1 Report

The authors have reported the effect of g-C3N4 addition into In2O3 for the H2 sensor. They found that the morphology of In2O3 was drastically changed to litchi-like by adding g-C3N4. G-C3N4 also improves the H2 sensing performance. The topics and results show in agreement with the focus of Sensors. Thus, the manuscript is suitable for publication. However, some of the critical data is lacking to explain their results. In addition, their proposed reasons for the improvement of sensing property are not reasonable and only speculation. Thus, after major revisions, I'd like to recommend this manuscript for publication in Sensors. My comments are below.     

Comment 1
The main text in 2.1., 2.2. and 2.4. is Italic. Please revise them.

Comment 2
The author claimed that g-C3N4 might inhibit the growth of In2O3. If so, the crystallite size also decreased as the amount of g-C3N4 increased. Please, calculate the crystallite size from XRD and discuss it in the revised manuscript.

Comment 3
Please add the label for the right-down image. (Maybe In?)

Comment 4
Why can you detect oxygen vacancy by XPS?  Oxygen vacancy means the absence of oxygen. Where did the electron come from?  In addition, OH always covers the surface of In2O3. Why did you ignore the surface OH? Did you perform the special treatment for removing the adsorbed OH? 

Comment 5
To clarify the C 1s spectrum, please add C 1s spectrum for pure In2O3. C always exists on the surface of the sample. In addition, I couldn't find the peak at 398.8 eV in the N 1s peak. If you discuss it, please extend the collection time to increase the S/N.

Comment 6
Why do your sensors exhibit the maximum sensor response at 275C? Please explain and discuss the reason in the revised manuscript.

Comment 7
Please insert the information about the H2 concentration in Fig 9 (a).

Comment 8
Please discuss the effect of g-C3N4 addition on the selectivity in the revised manuscript. Why does the response to only H2 increase? 

Comment 9

 I couldn't understand what they pointed out as the second reason for the g-C3N4 effect. Do you mean that the oxygen vacancy works as the adsorption site for O2-, O-, or O2-? Do you detect the chemisorbed oxygen at oxygen vacancy by XPS? If so, the explanation of the XPS spectrum assignment is incorrect. You should not use "oxygen vacancy; Ov" as the peak assignment around 531.2 eV for the O 1s spectrum. That is the chemisorbed oxygen at the surface oxygen vacancy. 

Furthermore, if you pointed out the chemisorbed oxygen at the surface oxygen vacancy, can O2- occupy the oxygen vacancy? It seems to be problematic from the viewpoint of molecular size.

 Additionally, you can discuss whether the second reason is correct when comparing the amount of chemisorbed oxygen from XPS for all samples. Please, show the O 1s spectrum for all samples and discuss it in the revised manuscript.

Why is the number of oxygen vacancies increased by adding the g-C3N4? Please explain it in the revised manuscript.

Comment 10
If the third reason is true, the resistance of the sensor should be increased as the amount of g-C3N4 increases.

Please, add the base resistance of each sensor and discuss it in the revised manuscript.

Reviewer 2 Report

The manuscript reported the highly selective gas sensor for H2 rapid hydrogen detection. However, there are some issues with the review that should be considered like: -

                  1.            The font style and size should be unique in the abstract. The same action is required in the whole manuscript.

                  2.            Authors should mention the purity and composition of the target gas in the experimental section.

                  3.            According to line number 123-125, it is not convincing. The XRD system Rigaku Miniflex 600 X can detect 1 mass% trace component. Please provide the amount of g-C3N4 and In2O3.

                  4.            All the references should be in the same pattern. In line number 307, the authors should properly write the compound with subscripts.

                  5.            Please cite the mentioned articles related to sensing materials; (i) Sensors and Actuators B: Chemical, 371, 2022, p.132504. https://doi.org/10.1016/j.snb.2022.132504, (ii) Journal of Electroanalytical Chemistry, 909, 2022, p.116115, (iii) Sensors and Actuators B: Chemical, 272, 2018, pp.100-109, (iv) Hussain, C.M., Thomas, S. (eds) Handbook of Polymer and Ceramic Nanotechnology. Springer, Cham., https://doi.org/10.1007/978-3-030-40513-7_27.

                  6.            The manuscript contains many grammatical errors and typos which should be corrected.

Round 2

Reviewer 1 Report

The author revised the manuscript according to the comments.